# Peroxide-Based Crosslinking of Solid Silicone Rubber, Part I: Insights into the Influence of Dicumylperoxide Concentration on the Curing Kinetics and Thermodynamics Determined by a Rheological Approach

**DOI:** 10.3390/polym14204404

**Published:** 2022-10-18

**Authors:** Maurício Azevedo, Anna-Maria Monks, Roman C. Kerschbaumer, Sandra Schlögl, Clemens Holzer

**Affiliations:** 1Polymer Competence Center Leoben GmbH, Roseggerstrasse 12, 8700 Leoben, Austria; 2Department of Polymer Engineering and Science, Polymer Processing, Montanuniversitaet Leoben, Otto Gloeckel-Strasse 2, 8700 Leoben, Austria

**Keywords:** silicone, dicumylperoxide, crosslinking kinetics, processing simulation, rheology, rubber process analyser

## Abstract

Predicting the curing behaviour of industrially employed elastomeric compounds under typical processing conditions in a reliable and scientifically driven way is important for rubber processing simulation routines, such as injection moulding. Herein, a rubber process analyser was employed to study the crosslinking kinetics of solid silicone rubber based on the concentration of dicumylperoxide. A model was proposed to describe the optimal cure time variation with peroxide concentration and temperature, based on the analysis of processing parameters applying kinetic and thermodynamic judgments. Additionally, the conversion rate was described with the aid of a phenomenological model, and the effect of dicumylperoxide concentration on the final crosslink state was investigated using kinetic and thermodynamic explanations. Optimal curing time was affected both by temperature and dicumylperoxide concentration. However, the effects were less pronounced for high temperatures (>170 ${}^{\circ }$C) and high concentrations (>0.70 phr). A limit on the crosslink state was detected, meaning that the dicumylperoxide capacity to crosslink the silicone network is restricted by the curing mechanism. Curing restrictions were presumed to be primarily thermodynamic, based on the proton abstraction mechanism that drives the crosslinking reaction. In addition to providing more realistic crosslinking models for rubber injection moulding simulation routines, the results of this study may also explain the chemical behaviour of organic peroxides widely used for silicone crosslinking.

## 1. Introduction

Poly(siloxane)s of high molecular weight, known as silicone rubbers, are materials based on macromolecules with successive silicon–oxygen bonds as a backbone, with the Si atom having two monovalent organic side groups. This alternating inorganic backbone of Si and O atoms provides strong and long chain bonds 106 kJ mol${}^{-1}$ and 1.64 Å, compared to 85 kJ mol${}^{-1}$ and 1.53 Å for C–C bond [1]), backbone flexibility (Si–O–Si bond angle of 143°, compared to 109.5° of the usual tetrahedron bond angles [2]), ease of side-group rotation, and low inter- and intramolecular forces [3]. As a result of the longer Si–O bond, compared to a C–C bond, there is more space for organic substituents, like methyl and phenyl, without an increase in steric hindrance or molecular congestion. Insertion of bulky substituents does not compromise the linearization of the molecules due to the long and flexible Si–O bond, allowing crystallization. The flexibility of the Si–O–Si bond is due to the oxygen atom delocalized lone electron pair, which is spread over the covalent bond region between Si and O atoms, specifically into the silicon vacant d orbital, allowing for a wider angle for the $sp^{3}$ hybridization. These molecular properties, linked to the inorganic nature of the backbone and the organic nature of the functional side groups, constitute an important bridge between inorganic and well-known organic polymers. Nowadays, silicone rubber has been used widely to produce medical devices, optical components, seals, and electronic parts, for example.

Silicone rubber, despite its very high molecular weight (>100,000 g mol${}^{-1}$), must be crosslinked to be employed in engineering applications [4]. The crosslinking reaction turns the soluble and highly viscous silicone rubber into an insoluble and viscoelastic network, following a specific crosslinking mechanism, kinetics, and thermodynamics. Industrially-applied elastomers, like the silicone rubber, are usually crosslinked by the use of specific chemicals (here denoted as crosslinkers), which along with temperature trigger the crosslinking reaction. Many are the chemical reactions that are able to join two adjacent macromolecules, being the most well-known ones the vulcanization (a mixed ionic-radical reaction at high temperature and applying sulphur, accelerators, and other additives) and the radical crosslinking (at high temperature usually employing organic peroxides) [5,6,7]. These two crosslinking methods are currently widely applied for producing rubber parts, employing different polymer processing techniques, such as compression and injection moulding. With the advent of numerical simulation, polymer processing has become a precise and highly technological industrial step. Before actual rubber injection moulding, for example, the process can be computer-aided simulated, in order to establish the most suitable processing conditions for a specific application [8,9]. During simulation, several models are applied to describe the injection moulding process, from the filling phase to the crosslinking phase of the product [10,11]. These models must account not only to describe the processing parameters, such as the time necessary to crosslink the silicone rubber product [12] or the pressure inside the cavity, but also to consider the kinetics and the thermodynamics of the related phenomena.

There are numerous scientific contributions in the field of silicone rubber crosslinking kinetics [13,14,15,16,17], like the one by Hernández-Ortiz and Osswald [18]; however, most of them do not approach processing-related parameters. Additionally, it is rare to find studies that cover the kinetics and the thermodynamics of silicone rubber crosslinking by analysing rheological experimental data. Thus, this study aims to cover these gaps in two different, but converging, fronts: first, to propose a processing-related point-of-view for the crosslinking kinetics by analysing important processing parameters; and second, to describe how the thermodynamics of dicumylperoxide-based silicone rubber crosslinking affects the overall crosslink state of the rubber product. The properties of the final silicone rubber network, which corroborate to the findings of the present study, are currently under investigation and will be thoroughly discussed in a separate publication.

## 2. Materials and Methods

This section briefly describes the elastomer under investigation and the methodology for the rheological measurements. For further details about the rubber process analyser device and its versatility, the following standards (ASTM, American Society for Testing and Materials, West Conshohocken, PA, USA) cover the main topics related to this kind of measurement: ASTM D5289 [19], ASTM D6204 (A, B, and C) [20], ASTM D6601 [21], and ASTM D8059 [22]. It is highly advised that the Appendix A to this publication is consulted for practical aspects of the described experiments.

### 2.1. Materials

High consistency poly(dimethylsiloxane), or solid PDMS (Xiameter™ RBB-2100-50, $M_{w}$ = 660 kg mol${}^{-1}$, $M_{w}/M_{n}$ = 1.8), containing approximately 26 wt% of an inorganic filler and no significant concentration of vinyl side groups (as confirmed by ${}^{1}$1H-NMR studies, see Appendix A) was supplied by Dow Inc. (Midland, MI, USA). The PDMS characterization in terms of molecular weight, filler, and vinyl contents is described in the Appendix A. Dicumylperoxide (DCP) 99.9% (Peroxan DC) was supplied by Pergan GmbH (Bocholt, Germany), with active oxygen content of 5.91 wt%. Both components were thoroughly mixed in a 2-roll-mill varying the peroxide concentration: 0, 0.21, 0.35, 0.49, 0.70, 1.00, and 1.50 phr of DCP. These concentrations represent fractions of the suggested peroxide concentration (0.70 phr) by the silicone supplier, i.e., 0, 30, 50, 75, 100, 140, and 215%, respectively. All silicone compounds were stored at low temperatures (<5 ${}^{\circ }$C) prior to testing, aiming to avoid premature pre-curing.

### 2.2. Rotational Rheometry

Rubber process analyser (D-RPA 3000 Montech Werkstoffprüfmaschinen GmbH, Buchen, Germany) equipment, or moving die rheometer (MDR), was employed as a rotational rheometer to characterize the curing behaviour of the PDMS/DCP compounds at different temperatures: 140, 150, 160, 170, and 180 ${}^{\circ }$C for 1 h. The rotational deformation was set to 0.5° at a constant frequency of 1.667 Hz. Processing-related parameters were calculated from the torque vs. time curves: minimum torque (ML), maximum torque (MH, as the torque value at t = 1 h, since no reversion or marching modulus was observed), scorch time ($t_{s1}$), and optimum cure time ($t_{90}$). The scorch time was defined as the time correspondent to increase ML by 1 dN m (torque unit), and the optimum cure time as the time correspondent to 90% of MH. All experiments were performed in triplicate for each sample (7 dicumylperoxide concentrations and 5 temperatures).

To describe the effect of temperature and dicumylperoxide concentration on the optimum cure time, fitting of the data was performed using the Levenberg–Marquardt algorithm [23] or damped least-squares method. A general equation to define $t_{90}$ was proposed, into which the experimental data was fitted:(1)$$t_{90}=A_{0}exp\left(\frac{E_{a}}{RT}\right){\left(DCP\right)}^{\beta }$$

In Equation (Equation 1), the optimum cure time is described as a function of two factors. To the temperature was assigned an Arrhenius relation, since time and temperature were under study. The influence of dicumylperoxide concentration, on the other hand, was described by a power law with order β.

For the crosslinking kinetics study, only the data related to the curing temperature of 160 ${}^{\circ }$C was used, since this is the common and the manufacturer-advised curing temperature for the silicone rubber under investigation. However, the usual modus operandi is to fit the data related to several thermal programs, those being either different temperatures or different heating rates. Three ASTM standards currently describe how to estimate kinetic parameters utilizing thermal analysis: ASTM E2041-13 (Standard test method for estimating kinetic parameters by differential scanning calorimeter using the Borchardt and Daniels method) [24], ASTM E698-18 (Standard Test Method for Kinetic Parameters for Thermally Unstable Materials Using Differential Scanning Calorimetry and the Flynn/Wall/Ozawa Method) [25], and ASTM E2781-11 (Standard Practice for Evaluation of Methods for Determination of Kinetic Parameters by Thermal Analysis) [26]. Along with these standards, the publications by Vyazovkin [27,28,29,30,31] give important information related to determining the kinetics of several phenomena, such as crystallization, glass transition, and crosslinking.

## 3. Theoretical Background

The description of curing kinetics in terms of experimental values is detailed below by applying a phenomenological model. In contrast to the mechanistic models, which are based on solving numerous differential equations that describe mass action kinetics, these are formulated based on the curing degree, or conversion [32]. For engineering and processing simulation purposes, phenomenological models produce parameters that are easy to implement, making them more convenient than mechanistic models. As a result, the phenomenological model considers the curing reaction as a single chemical step, which is obviously a simplification of the whole crosslinking phenomenon that includes several sequential reactions, which will be discussed in the following section.

### 3.1. Crosslinking Kinetic Modelling

Concerning rheological experiments, the crosslink conversion rate $\frac{d\alpha }{dt}$ can be calculated considering the difference between the torque before and after the curing reaction and the actual torque at a certain time *t*. Specifically for RPA, the torque before curing is defined as the minimum torque (ML) and the torque after curing as the maximum torque (MH), being their difference expressed as ΔM=MH−ML. Using Equation (Equation 2), the conversion rate at a given point in time can be calculated based on the following mathematical relation:(2)$$\frac{d\alpha }{dt}=\frac{1}{\Delta M}\frac{d(M-ML)}{dt}$$

Considering that crosslink formation is a temperature-induced process, the same conversion rate can be represented as a function of temperature k(T), and as a function of the appropriate kinetic model f(α), as reviewed by Vyazovkin [28]:(3)$$\frac{d\alpha }{dt}=k\left(T\right)f\left(\alpha \right)$$

It is well-known and reported that k(T) is an Arrhenius-like rate constant, i.e., it is a correlation between temperature *T* (K), a pre-exponential factor *A* (time${}^{-1}$), an activation energy *E* (J mol${}^{-1}$), and the molar gas constant *R* (8.3145 J mol${}^{-1}$ K${}^{-1}$) as follows:(4)$$k\left(T\right)=Aexp\left(\frac{E}{RT}\right)$$

Substituting Equation (Equation 4) into Equation (Equation 3), and taking a logarithm followed by differentiation against $\frac{1}{T}$ gives rise to the equation that defines the isoconversional principle: considering two samples, if the curing rates $\frac{d\alpha }{dt}$ are the same and the conversions α are equal, then the curing rate is only a function of temperature. The isoconversional principal equation is then written as:(5)$${\left[\frac{\partial \ln(\frac{d\alpha }{dt})}{\partial T^{-1}}\right]}_{\alpha }={\left[\frac{\partial \ln(A)}{\partial T^{-1}}\right]}_{\alpha }+{\left[\frac{\partial \ln(f(\alpha ))}{\partial T^{-1}}\right]}_{\alpha }-{\left[\frac{E}{R}\right]}_{\alpha }$$

Analysing Equation (Equation 5) enables one to make the following conclusions: the term ${\left[\frac{\partial ln(A)}{\partial T^{-1}}\right]}_{\alpha }$ is zero, since differentiating a constant results zero; and ${\left[\frac{\partial ln(f(\alpha ))}{\partial T^{-1}}\right]}_{\alpha }$ also equals zero, since the kinetic model f(α) is only a function of the conversion. Finally, one can conclude that the activation energy for a certain conversion value α (now referred as $E_{\alpha }$) does not depend on the chosen kinetic model, i.e., it is model-free:(6)$$E_{\alpha }=-R{\left[\frac{\partial \ln(\frac{d\alpha }{dt})}{\partial T^{-1}}\right]}_{\alpha }$$

The assumption that the kinetic model does not change with temperature is reasonable, since the temperature range related to a given conversion is so narrow that no change would be detected [28]. Obtaining the values for activation energy in terms of conversion without taking the kinetic model into consideration is very convenient and already gives light to important details about the crosslinking mechanism. Thus, this principle was employed to study the crosslinking kinetics during the present study.

There are many different approaches that apply the isoconversional principle to determine the activation energy of thermally induced processes. Vyazovkin et al. [27] and Zhang [33] give a broad overview about these methods, which include the integral isoconversional one used in this research: the Friedman-like [34], for isothermal experiments, represented by Equation (Equation 7).
(7)$$\ln\left(t_{\alpha ,i}\right)=\ln\left[f\left(\alpha \right)A_{\alpha }\right]-\frac{E_{\alpha }}{RT_{\alpha ,i}}$$

The subscripts α,i indicate a given conversion α and a given thermal program *i* (temperature for isothermal experiments or heating rate for dynamic experiments) [29]. It is possible to notice after careful analysis that Equation (Equation 7) can be written in the linear form y=ax+b, assuming $\frac{1}{T}$ as *x*. Thus, the activation energy $E_{\alpha }$ can be determined for each conversion value (here represented by $t_{\alpha ,i})$ by plotting the left-hand side of Equation (Equation 7) over $\frac{1}{T}$. This strategy was used in the present research to determine an initial guess for the activation energy, to be implemented in the subsequent fitting procedure.

Regarding the kinetic model function f(α), it is widely accepted [13,14,16,28] that the crosslinking reaction of silicone rubbers follows the autocatalytic model, first proposed by Šesták-Berggren [35] in the form of Equation (Equation 8), and further modified by Kamal [36] to include the temperature dependence, as seen in Equation (Equation 9):(8)$$f\left(\alpha \right)=\alpha^{m}{\left(1-\alpha \right)}^{n}$$
(9)$$\frac{d\alpha }{dt}=\left(k_{1}+k_{2}\alpha^{m}\right){\left(1-\alpha \right)}^{n}$$
where *m* and *n* are the reaction orders and $k_{1}$ and $k_{2}$ are the Arrhenius rate constants, as previously defined in Equation (Equation 4). For the present study, a third version of the autocatalytic model was assumed, where $k_{1}$ and $k_{2}$ are taken as constants with activation energies $E_{\alpha ,1}$ and $E_{\alpha ,2}$, in a way that the final kinetic model equation can be written, after proper substitution using Equations (Equation 4) and (Equation 9), as:(10)$$\frac{d\alpha }{dt}=\left[A_{1}exp\left(\frac{-E_{\alpha ,1}}{RT}\right)+A_{2}exp\left(\frac{-E_{\alpha ,2}}{RT}\right)\alpha^{m}\right]{\left(1-\alpha \right)}^{n}$$

The subscript 1 denotes the $n^{th}$ order contribution to the crosslinking model, while the subscript 2 denotes the auto-catalytic contribution. Similarly, *n* is the reaction order for the $n^{th}$ model, and *m* is the reaction order for the auto-catalytic part.

The RPA data (calculated conversion rate $\frac{d\alpha }{dt}$ values based on experimentally-determined torque data, according to Equation (Equation 2)) for T = 160 ${}^{\circ }$C was fitted to Equation (Equation 10) for conversion α values between 0.1 and 0.9, as normally recommended [13,27]. Initially, it was assumed [13] that $E_{\alpha ,1}=E_{\alpha ,2}=E_{\alpha }$ as calculated applying the isoconversional approach. This $E_{\alpha }$ value was then implemented as the activation energy parameters initial guess at the preliminary fitting routine and the parameters $A_{1}$, $A_{2}$, *m*, *n*, and $E_{\alpha }$ were determined employing the Levenberg–Marquardt algorithm [23] or damped least-squares method. After the first preliminary fitting, the previously calculated parameters were employed as initial guesses for the final calculations of the kinetic parameters $A_{1}$, $A_{2}$, *m*, *n*, $E_{\alpha ,1}$, and $E_{\alpha ,2}$. All fitting procedures were performed utilizing Python coding language (Python 3.8, PyCharm Community Edition 2021.2.2, Jet Brains, Prague, Czechia).

### 3.2. Peroxide-Based Crosslinking Mechanism of PDMS

Crosslinks formation between two adjacent PDMS macromolecules, when aided by an organic peroxide, is based on the generation of radicals by the peroxide decomposition reaction. The interaction between peroxide radicals and PDMS macromolecules can occur via hydrogen abstraction from a methyl and/or vinyl groups or via addition to the double bond on the vinyl group, if this is present on the chain. If the energies of the resulting radicals are compared and considering that hydrogen abstraction is more favourable if the radical energy is lowered, hydrogen abstraction from the vinyl group is unfavourable, due to the similar energy of alkoxy and vinylic radicals [37].

When dicumylperoxide is concerned, the main chemical reactions involved during PDMS crosslinking via hydrogen abstraction from methyl side groups are:i$C_{18}H_{22}O_{2}\overset{\Delta }{\to }2C_{9}H_{11}O\;\cdot \;$ii$C_{9}H_{11}O\;\cdot \;⟶C_{8}H_{8}O+CH_{3}\;\cdot \;$iii$C_{9}H_{11}O\;\cdot \;+\;[Si$–*O*–$Si\left(CH_{3}\right){]}_{n}⟶C_{9}H_{11}OH+[Si$–*O*–$Si\left(CH_{2}\;\cdot \;\right){]}_{n}$iv$CH_{3}\;\cdot \;+\;[Si$–*O*–$Si\left(CH_{3}\right){]}_{n}⟶CH_{4}+[Si$–*O*–$Si\left(CH_{2}\;\cdot \;\right){]}_{n}$v2[Si–*O*–$Si\left(CH_{2}\;\cdot \;\right){]}_{n}⟶[Si$–*O*–$Si(CH_{2})$–$(CH_{2})Si$–*O*–${Si]}_{n}$vi$C_{9}H_{11}OH⟶C_{9}H_{10}+H_{2}O$

Radical generation is accomplished at reactions i and ii, via thermal decomposition of dicumylperoxide and beta scission of the cumyloxy radical, generated in the reaction i, respectively. There are two radicals responsible for silicone crosslinking: the cumyloxy and the methyl radicals, which are able to abstract a proton from the methyl side group of PDMS (reactions iii and iv), creating two macromolecular radicals. It was shown by Baquey et al. [38] that the main responsible for PDMS proton abstraction is the radical directly formed after peroxide decomposition, which in the present case is the cumyloxy. However, the methyl radical is also considered in this study as able to abstract protons from the silicone polymer chain. Finally, the macromolecular radicals, when adjacent to each other, can react with one another, creating a covalent bond between the carbon atoms from the methyl side groups (reaction v).

For the case where double bond addition occurs, the cumyloxy (denoted as ***R*** and the methyl (denoted as ***R’***) radicals can add to the less-substituted secondary carbon (due to its higher stability as intermediate radical) of the vinyl group, as shown in reactions vii and viii, leading to the formation of macromolecular radicals. Similar to what happens in the hydrogen abstraction case, adjacent macromolecular radicals are able to react with each other by recombination, creating a crosslink point (reaction ix), or the macromolecular radical can continue adding to another adjacent vinyl group. It is worth mentioning that additional reactions result in products with fragments from the peroxide initiator, i.e., the fragment ***R*** or ***R’*** is attached to the polymer chain.

viiR·+[Si–*O*–$Si(H_{2}C$=${CH)]}_{n}⟶C_{9}H_{12}O+[Si$–*O*–$Si(H_{2}CR$–${CH\;\cdot \;)]}_{n}$viii$R^{\prime }\;\cdot \;+\;[Si$–*O*–$Si(H_{2}C$=${CH)]}_{n}⟶C_{9}H_{12}O+[Si$–*O*–$Si(H_{2}CR^{\prime }$–${CH\;\cdot \;)]}_{n}$viiii2[Si–*O*–$Si(H_{2}CR$–${CH\;\cdot \;)]}_{n}$⟶$[SiOSi\left(H_{2}CRCH\right)$–$\left(CHRCH_{2}\right){SiOSi]}_{n}$

When vinyl moieties are present in the PDMS backbone, hydrogen abstraction and vinyl addition happen simultaneously, even though, considering alkoxy (from dicumylperoxide) and methyl radicals (both 105 kcal mol${}^{-1}$), the formation of secondary $R_{2}CH\;\cdot \;$ radicals is more favourable than primary $RCH_{2}\;\cdot \;$ radicals (97 against 100 kcal mol${}^{-1}$, respectively) [37], due to the alkyl substitution stability effect. However, vinyl addition extension is limited to the vinyl concentration on the PDMS chain, while hydrogen abstraction has virtually no concentration limitation in terms of methyl presence; it is only limited due to steric hindrance and/or thermodynamical reasons.

Concurrently to dicumylperoxide-mediated crosslink formation in PDMS, by-products are formed due to hydrogen abstraction or decomposition of radicals. In reaction iii, cumyl alcohol $C_{9}H_{11}OH$ is formed after hydrogen abstraction by the cumyloxy radical, and further dehydrated and converted into α-methylstyrene (reaction vi). In reaction ii, acetophenone $C_{8}H_{8}O$ is the by-product of cumyloxy beta scission. All these by-products do not contribute to the network formation, but are treated and understood as low molecular weight and volatile compounds that may migrate, causing mainly the very characteristic odour of peroxide-crosslinked silicone products. Cumene may also be formed after reduction of α-methylstyrene [39].

The aforementioned crosslinking kinetic model accounts for the overall crosslink formation. This model is not able to differentiate between abstraction or addition reactions due to its phenomenological approach [40], but clarifies the rate of crosslink formation. A complete description of the complex set of reactions that underline peroxide crosslinking of PDMS can be performed considering mechanistic approaches, like population balance equations, which is not the main focus of the present study, where a phenomenological kinetic model was proven to be sufficient.

## 4. Results

This section is divided into three major subsections. The first, comprehending Section 4.1, discusses the curing kinetics and its association with processing conditions, proposing a mathematical relation for the optimum cure time based on the experimental data. The second, including Section 4.2, reports the crosslinking kinetics and presents the kinetic parameters, along with the reasoning behind their trend as the dicumylperoxide concentration increases. Finally, the third, Section 4.3, presents thermodynamic considerations regarding the dicumylperoxide-based crosslinking of silicone rubber.

### 4.1. Curing Characteristics

Figure 1 displays the curing curves for the PDMS/DCP systems for different dicumylperoxide concentrations and at different curing temperatures. Curing curves are usually analysed in rubber industry and research to identify the crosslinking profile in terms of minimum detected torque, induction time, velocity, curing time, and maximum detected torque. A torque increase is present for all samples with dicumylperoxide, regardless of the concentration, due to the formation of a denser polymer network, promoted by crosslinking. Connecting adjacent PDMS macromolecules forms a stiffer and more elastic network, which imposes a higher torque to the device, in order to keep the set deformation.

For all dicumylperoxide concentrations, the temperature increase shortens the induction or scorch time and lowers the minimum detected torque, as also depicted in Table 1. Since dicumylperoxide decomposition is initiated and promoted by heat (reaction i in Section 3.2), a higher curing temperature delivers more heat to the rubber compound, leading to a shorter crosslinking induction time, represented by the $t_{s1}$ values. For a constant curing temperature, the higher the dicumylperoxide concentration, the lower the induction time. The effect of dicumylperoxide concentration on the induction time is explained by the higher concentration of radicals that are formed after decomposition and by the autocatalytic nature of this process, as also reported by Kruželák et al. [41,42]. Industrially, this value is related to process safety and connected to the mouldability of the rubber compound, which is limited after $t_{s1}$ is reached. As a matter of example, short scorch times are preferred for extrusion of hollow rubber profiles, to avoid extruded collapse after passing through the matrix. On the other hand, for rubber injection moulding, scorch time must be long enough to guarantee the complete filling of the mould before curing.

The minimum detected torque ML was also slightly lowered by temperature increase, which is mainly an effect of temperature over the rubber compound’s viscosity. The effect of temperature over rubber viscosity is well known, and the processability is related to this value when RPA is employed in the rubber industry. Along with the values for Mooney viscosity, the minimum detected torque can clarify aspects about rubber processing, including mixing and moulding. No effects of dicumylperoxide concentration were detected at the minimum torque due to the low concentration, the low molecular weight, and the dispersion of the peroxide into the rubber matrix.

The curing rate is enhanced by temperature, as the cure curves, for all dicumylperoxide concentrations, become steeper for higher curing temperatures. Dicumylperoxide decomposition kinetics is favoured by temperature [43,44], leading to a faster crosslinking reaction for the PDMS/DCP systems as well. Crosslinking velocity is also promoted by higher concentrations of dicumylperoxide, due to the higher concentration of radicals and the autocatalytic nature of this process. Table 2 describes the optimal cure time $t_{90}$, which decreases as temperature and dicumylperoxide concentration increase. This decrease is an effect of temperature over the dicumylperoxide decomposition and rubber crosslinking kinetics, and of the quantity of radicals that are formed. Industrially, the selection of the most appropriate curing time is a compromise between processing time, processing temperature, and part quality in terms of mouldability.

As an indicative of the final crosslink density, the maximum detected torque represents the maximum shear force resistance of the rubber compound at a given time and temperature, for the set deformation [4]. The maximum torque values, as shown in Table 2, tend to increase with temperature and with the dicumylperoxide concentration, suggesting that the effective number of crosslink points is also enhanced by temperature and dicumylperoxide amount. Even though the maximum torque values increase with dicumylperoxide, a threshold of approximately 11–12 dN m is reached for the highest temperatures/dicumylperoxide concentrations.

Regarding the samples without dicumylperoxide, an increase in the detected torque is observed for all samples, with a steeper increase for higher temperatures. Even though a decrease in the torque within the first minutes due to viscosity reduction is detected, torque increases with time, which may indicate some crosslinking phenomenon not associated to the presence of dicumylperoxide, but also triggered by temperature. This phenomenon may be associated to peroxide or metal contamination, that ultimately would lead to crosslinking when the temperature is increased.

When cycle time for moulding is evaluated for industrial applications, the optimum cure time $t_{90}$ is a factor usually taken into account. An estimation for the optimum cure time within the temperature and dicumylperoxide ranges that were investigated is shown in Figure 2, taking Equation (Equation 1) as the mathematical model for the parameter. The proposed equation can fit almost 99% of the experimental data, as represented by the $R^{2}$ factor.

Temperature has a major influence over the optimum cure time for lower peroxide concentrations, while for concentrations above 0.70 phr the effect is not prominent. Similarly, the peroxide concentration has a stronger influence on the optimum cure time for lower temperatures, becoming weaker as the temperature increases. The less distinguished effect on the optimum cure time for higher temperatures and higher DCP concentrations is related to two distinguished reasons. For the temperature, the higher range (>150 ${}^{\circ }$C) is well above the dicumylperoxide decomposition temperature, which is around 120 ${}^{\circ }$C [44]. This means that the decomposition and the preceding crosslinking reaction will take place almost instantly and rapidly, and the different peroxide concentrations will not impact the optimum cure time in a notable way. Regarding the DCP concentration, the higher concentration range (>0.70 phr) shows a weaker effect of temperature on the optimum cure time, possibly due to the autocatalytic nature of peroxide-based crosslinking reactions. In these cases, the number of generated radicals is enough to compensate the temperature effect and overlap it. Hou et al. [45] recently reported the same effect of temperature on the optimum cure time for dicumylperoxide-crosslinked silicone rubber.

Here in this paper, an equation is proposed to model the optimum cure time for peroxide-cured high consistency silicone elastomers. The parameters for this equation are also described in Figure 2. The activation energy is associated to the amount of energy necessary to reach the optimum cure time, which is also associated to the amount of energy necessary to decompose the dicumylperoxide molecules as stated in the chemical equation i in Section 3.2. Thus, the value of 115 kJ mol${}^{-1}$ is in accordance with the values reported in the literature [43,44] for DCP decomposition, and with the activation energy values that will be presented further in the crosslinking kinetics discussion. The exponent for the dicumylperoxide concentration has a negative value due to the decrease of the optimum cure time with the peroxide concentration and is less than 1 possibly due to its limitation in crosslinking the polymer, which will also be further explained. The pre-exponential factor, similarly to the activation energy, presents a value that will also similarly reported when the crosslinking kinetics is discussed further in this paper. It is important to state here that these parameters and this specific equation fit to the polymer and to the crosslinking system that were investigated in this paper, at the same time. For other polymeric systems, with different crosslinking mechanisms and kinetics, another set of equations and parameters must be proposed and evaluated.

### 4.2. Crosslinking Kinetics

Prediction of how a polymer crosslinks when exposed to heat during typical processing conditions, like compression or injection moulding, is a key step to control the overall process and the part quality. The crosslinking kinetics, i.e., the evolution of conversion α over time, was evaluated in terms of the torque increase detected at the rubber process analyser at 160 ${}^{\circ }$C, which was the temperature set for compression moulding.

Based on the curves shown in Figure 1, the conversion rate $\frac{d\alpha }{dt}$ was calculated employing Equation (Equation 2) and is plotted versus time in Figure 3a and versus conversion in Figure 3b. The effect of increasing the dicumylperoxide concentration is seen as an acceleration of the reaction, i.e., higher cure velocity, when analysing the conversion rate evolution on time. Additionally, the maximum cure rate is shifted to lower times. These findings are related to the concentration of peroxide radicals that are formed at 160 ${}^{\circ }$C and its catalytic effect on the formation of further radical species. Regarding the evolution of cure velocity over conversion, or reaction path, the maximum cure rate remains constant at a fixed conversion value between 0.2 and 0.4 for the dicumylperoxide concentrations up to 0.70 phr. This means that the highest rate of crosslink formation occurs at a constant conversion range, regardless of the dicumylperoxide concentration below 0.7 phr. It is possible to speculate that the formation of the characteristic three-dimensional crosslinked network occurs mostly within the 0.2 and 0.4 conversion range for DCP concentration equal or lower than 0.70 phr, as a result of the decomposition of the majority of dicumylperoxide molecules. Within this conversion range, the concentration of radicals likely decreases, either by transfer to the polymer chain or by termination, resulting in the end of the crosslinking reaction. For the samples with higher dicumyperoxide concentrations (1.00 phr and 1.50 phr), the abruptly fast crosslinking reaction, as seen by the conversion rate values, shifts the maximum reaction speed to lower conversion values, probably due to the high amount of radicals that are formed.

As extensively described by Vyazovkin et al. [27,28,31], the determination of the kinetic parameters starts with calculating the activation energy without defining the most appropriate kinetic model, a strategy that is called model-free. The activation energy calculation following a Friedman-like approach as described by Equation (Equation 7) is exemplified for the 0.21 phr sample in Figure 4a. The slopes of the linear-fitted lines (correlation factor for the linear fittings were higher than 0.960) were used to determine the activation energy variation with conversion, as shown in the diagram of Figure 4b. A decrease of the activation energy is realized as the dicumylperoxide content increases, representing thermodynamically easier crosslinking reactions. This behaviour was already observed before, but in a kinetic way: the higher peroxide concentration accelerated the crosslinking reaction. Both thermodynamic and kinetic phenomena are behind the auto-catalytic nature of dicumylperoxide decomposition, which precedes the polymer crosslinking. Bianchi et al. [46] observed the same behaviour when studying the crosslinking of poly(ethylene-vinyl acetate) with dicumylperoxide, similarly reporting activation energy values between 85 and 105 kJ mol${}^{-1}$.

Duh et al. [43] reported an activation energy for dicumylperoxide thermal decomposition of 110–150 kJ mol${}^{-1}$, while Lv et al. [44] reported values between 150 and 200 kJ mol${}^{-1}$, which agree with the activation energy determined in the present study, i.e., varying between 85 and 140 kJ mol${}^{-1}$.

The activation energy is fairly constant in the conversion range 0.1–0.9, which is reasonable for a system that does not undergo drastic molecular diffusion changes during crosslinking, which would impose mass transfer limitations to the process [28]. Systems that experience a greater change in viscosity, like epoxy thermoset resins, are more sensitive to the diffusion hindrance caused by the crosslinking reaction (due to increase in viscosity), representing an increase of the activation energy [47]. Another consideration that deserves commenting is the heat transfer change due to the formation of a crosslinked layer at the top and bottom of the specimen inside the rubber process analyser during the first instances of curing. As studied by Cheheb et al. [48], the thermal conductivity of a crosslinked rubber sample is 10% higher than a non-crosslinked one, but still much lower than the metallic RPA cavity. This means that during curing, the heat transfer changes. However, this limitation was not further studied in the present work due to the low thickness of the RPA specimen (which imposes a fast heating of the whole sample) and the lack of scope of this investigation. Since the activation energy exhibited a relatively constant trend with conversion, the experimental data were fitted to the Kamal model using the mean values to calculate the kinetic parameters. Activation energy deviations that are observed at higher conversion values (α>0.9), which may indicate a decrease of the activation energy, arise from the small variation of torque at the final stages of crosslinking, which leads to high standard deviation of data.

Conversion rate data were fitted applying Equation (Equation 10), resulting in the kinetic parameters that are shown in Table 3. The calculated kinetic energies are in accordance with the values previously determined via the Friedman approach, and also showing a decrease with the dicumylperoxide concentration. Reaction orders *m* and *n* do not follow a specific trend and are considered here as being rather constant. Considerations regarding the reaction orders are associated with the crosslinking reaction mechanism, which is understood as the same regardless of the peroxide’s quantity. The pre-exponential values $A_{1}$ and $A_{2}$ have a tendency of decrease as the dicumylperoxide concentration increases.

However, analysing the pre-exponential factor alone may lead to misunderstandings [31], so the Arrhenius rate constants $k_{1}$ and $k_{2}$ are also reported in Table 3. The constants $k_{1}$ and $k_{2}$ are measurements of the system’s reactivity and are enhanced as the dicumylperoxide concentration increases. This is an additional evidence that corroborates the catalytic effect of the dicumylperoxide concentration.

Analysing the pre-exponential factors solely may lead to misinterpretations since it is often related to the activation entropy, meaning that the higher the pre-exponential factor, the more accelerated the chemical reaction is, as stated in Equation (Equation 11) [49]. This mathematical relation is a different way of expressing the Arrhenius rate constant previously denoted as Equation (Equation 4).
(11)$$k\left(T\right)=\frac{k_{B}T}{h}exp\left(\frac{\Delta S}{R}\right)exp\left(\frac{-\Delta H}{RT}\right)$$

The factor that contains the activation entropy ΔS, the Boltzmann constant $k_{B}$, and the Planck constant *h* was previously assigned as the pre-exponential factor *A*; while the inverse dependence on temperature is represented by the activation enthalpy ΔH factor. In the case of the present PDMS/DCP systems, it is not correct to assume that due to the decrease of the pre-exponential factor, the crosslinking reaction is decelerated as the dicumylperoxide concentration increases. Actually, the effect on the system’s reactivity, or the overall free energy of the activated reaction intermediate ΔG, is dominated by the enthalpy factor ΔH via the decrease of the activation energy, which is a pure effect of the peroxide catalyst’s nature, as explained by the usual thermodynamic relationship, derived from the second law of thermodynamics:(12)ΔG=ΔH−TΔS

The free energy ΔG is thus lowered strongly by the decrease of the activation enthalpy ΔH in comparison with the not so evident entropic effect. It is beyond the scope of this paper to explain why the activation entropy decreases as the dicumylperoxide increases, thus justifying further investigations. However, one rationalization may be that the active sites for radical reaction, which ultimately leads to a crosslink point, become limited as the dicumylperoxide concentration increases, possibly justifying the crosslinking threshold that is observed on the final crosslinked elastomer specimen. A second reasoning is that as the dicumylperoxide concentration increases, the radical concentration is also enhanced, increasing the probability of radical recombination before the reaction with the polymer chain, likewise limiting the crosslinking process. The last reasoning was already studied by Parks and Lorenz [50] when investigating the efficiency of dicumylperoxide to react with dimethyloctadiene.

After time integration, the conversion values calculated according to the model parameters were compared to the experimental ones, which are plotted in Figure 5. Within the conversion range 0.1–0.9, the calculated conversion values accordingly predict the experimental values with correlation factors higher than 0.99. For the specific temperature (160 ${}^{\circ }$C) and the dicumylperoxide concentrations that were evaluated, it is possible to predict how the crosslinking conversion behaves with time, meaning that this model can be possibly integrated into processing simulation routines. Along with the proposed mathematical relationship depicted at Equation Equation 1 for the optimum cure time, these suggestions may strongly enhance the quality of processing simulations that deal with reactive systems, such as elastomers.

Incorporating the variable dicumylperoxide concentration into the crosslinking kinetics model was not included in the scope of this paper, since it would demand an effort beyond the selected scientific targets. However, the ideal crosslinking kinetics model should embody not only the temperature of curing, but also a factor that correlates with the elastomer recipe in terms of the crosslinking systems. This becomes a suggestion for further studies.

### 4.3. Thermodynamic Reasoning behind the Dicumylperoxide-Based Crosslinking Reaction

Since no significant vinyl concentration was detected at the PDMS matrix via high-field solution ${}^{1}$H-NMR (as described in the Appendix A), considerations regarding the crosslinking mechanism relies on analysing the radical energies associated to the involved species. As reported by Dluzneski [37], the radical ability to abstract protons obeys the following ranking, where the bond dissociation energy of the first proton abstraction is expressed inside parenthesis and in kcal mol${}^{-1}$ (Ph stands for $C_{6}H_{5}$):


$$Ph–O\;\cdot \;(87)<Ph–C\;\cdot \;(88)∼C=C–C\;\cdot \;\left(88\right)<R_{3}C\;\cdot \;\left(91\right)<R_{2}HC\;\cdot \;\left(97\right)<\;RH_{2}C\;\cdot \;\left(100\right)<C=C\;\cdot \;\left(104\right)<H_{3}C\;\cdot \;\left(105\right)∼R–O\;\cdot \;(105)$$


*R*–O· and $H_{3}C\;\cdot \;$ radicals are generated with the higher bond dissociation energies, meaning they are the ones more prone to abstract a proton from the organic side group at the polymer chain. Considering reactions iii and iv as previously described in Section 3.2, these are characterized by the radical stabilization energy (RSE) [51] of the newly formed radical, which is the macromolecular radical [Si–*O*–$Si\left(CH_{2}\;\cdot \;\right){]}_{n}$, relative to the methyl and cumyloxy radicals. A macromolecular radical of this type behaves like a secondary carbon radical $RH_{2}C\;\cdot \;$, where R is the polymer backbone, meaning it has energy of about 100 kcal mol${}^{-1}$. For the proton abstraction reaction to happen, the RSE has to be negative, i.e., a decrease in the radical energy has to occur. For the related radical species, a decrease of only 5 kcal mol${}^{-1}$ takes place, which is probably enough for the reaction to thermodynamically proceed. This net decrease of around 5 kcal mol${}^{-1}$ is confirmed by an analogy by Bordwell et al. [52], where they report that the RSE for $Me_{3}SiCH_{2}\;\cdot \;$ when compared to the methyl radical $H_{3}C\;\cdot \;$ is −6 kcal mol${}^{-1}$.

Furthermore, this thermodynamic aspect related to the decrease in the radical energy can be understood as an enthalpic impediment, different from an entropic boundary that rises if steric hindrance or diffusion limitations takes place. By analysing once again Equation (Equation 12), one can realize that the factor affecting the reaction free energy is the enthalpic contribution, by decreasing the activation energy. This thermodynamic “boost” is associated to the crosslinking reaction and accomplished by the catalytic effect, i.e., decreasing the activation energy of the crosslinking process and allowing the reaction to take place in a thermodynamically favoured scenario, given the thermodynamic circumstances of proton abstraction. Besides, increasing the temperature also enhances the entropic factor −TΔS, which also leads to a decrease of the free energy.

Thus, one can understand that the crosslinking reaction is limited by the proton abstraction capacity of the radical species *R*–O· and $H_{3}C\;\cdot \;$, which, even though it exists, is considered as having a low thermodynamic impulse (RSE∼5 kcal mol${}^{-1}$). This is perhaps the factor that impedes the dicumylperoxide concentration to improve the crosslink density, as detected by the limited maximum detected torque. One evidence that corroborates this reasoning is the fact that Verheyn et al. [53] reported a substantial increase of silicone rebound resilience as the concentration of either di(2,4-dichloro-benzoyl)peroxide or di(4-methylbenzoyl)peroxide increased. Both peroxides decompose, producing aryl carboxylic radicals, which are even more stable than cumyloxy and methyl radicals, leading to more negative values for RSE.

## 5. Conclusions

This work reported a processing-oriented crosslinking kinetics study of a solid silicone rubber, or high consistency silicone, which is a widely employed material for manufacturing of technical rubber products. The investigation was carried out applying a rheological approach, by employing an industrially well-known device, the rubber process analyser (RPA). Different dicumylperoxide concentrations were studied, in order to determine their effect on processing parameters and on kinetic parameters.

Rheological experimental data showed that dicumylperoxide concentration enhancement accelerates the crosslinking reaction by shortening the induction time and increasing the reaction velocity. Besides, it results in higher but limited detected maximum torque, with distinguished extents for lower and higher temperatures. A correlation between the optimum cure time, the curing temperature, and the dicumylperoxide concentration was given, aiming to predict the cycle time during compression or injection moulding, for example.

Determination of the kinetics parameters showed that even though there was a decrease of the pre-exponential factor, the overall reactivity (represented by the Arrhenius rate constant) increased as the dicumylperoxide concentration rose. The overall reactivity was boosted due to the activation energy decrease caused by the increase on the dicumylperoxide concentration. This effect was associated to the catalytic effect of the radicals responsible for crosslinking.

For the first time, a thermodynamic and kinetic-based explanation was given for the limited effect of dicumylperoxide crosslinking efficiency on solid poly(dimethylsiloxane) rubber, considering the hydrogen abstraction mechanism. The findings of this work are useful for simulating processing cycles, enhancing the simulation similarity with real moulding. The effect of this specific crosslinking mechanism on the final silicone network and resulting properties is currently under investigation and will be disclosed in a separate study.

## Figures and Tables

**Figure 1 polymers-14-04404-f001:**
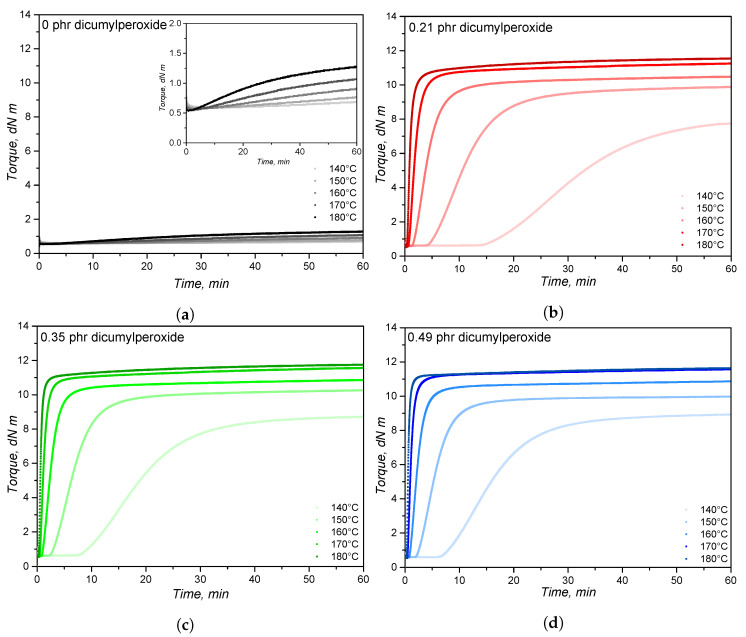
Rotational rheometry (RPA, 0.5° deformation at 1.667 Hz) curves for silicone rubber compounds with varied dicumylperoxide concentrations (**b**–**g**) at different crosslinking temperatures. The plot (**a**) (torque vs. time curve) is related to the pure solid silicone rubber without dicumylperoxide.

**Figure 2 polymers-14-04404-f002:**
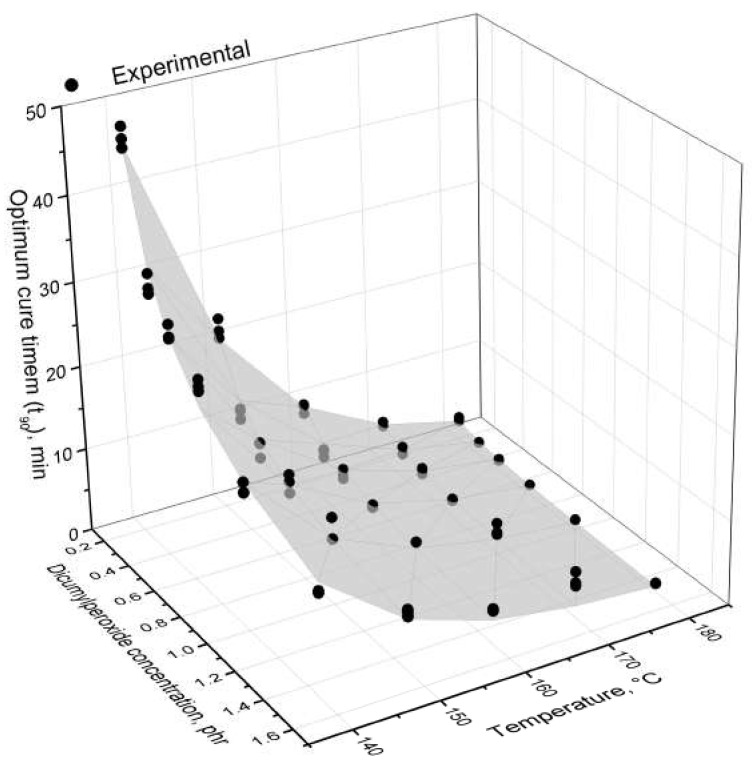
Optimum cure time $t_{90}$ as a function of the dicumylperoxide concentration and the temperature. The dots represent the experimental data and the surface indicates the model described at Equation (Equation 1), with fitting parameters as follows: $A_{0}$ (pre-exponential factor) = 2.02 × 10${}^{-14}$ min${}^{-1}$, $E_{a}$ (activation energy) = 1.182 × 10${}^{5}$ J mol${}^{-1}$, and β = —0.603. The correlation factor R${}^{2}$ was 0.988.

**Figure 3 polymers-14-04404-f003:**
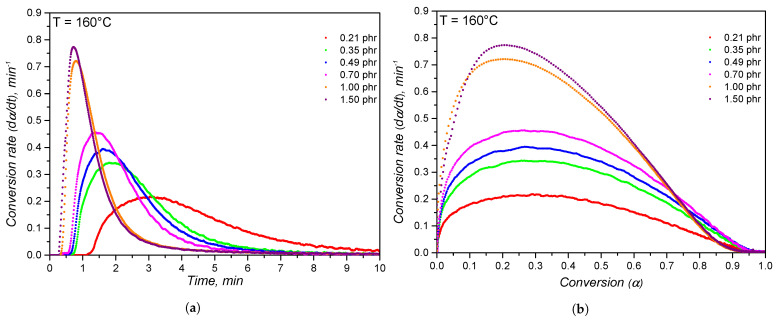
Conversion rate $\frac{d\alpha }{dt}$ at 160 ${}^{\circ }$C for silicone rubber compounds with different dicumylperoxide concentrations as a function of time (**a**) and conversion (**b**).

**Figure 4 polymers-14-04404-f004:**
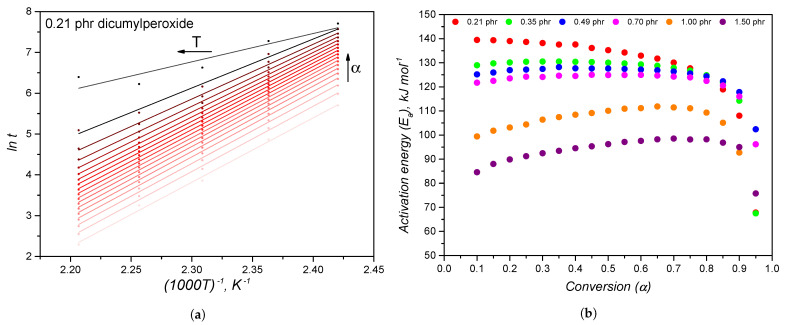
Friedman-like isoconversional approach for calculating the activation energy, regarding the sample 0.21 phr (**a**) and the activation energy for different dicumyperoxide concentration as function of conversion (**b**).

**Figure 5 polymers-14-04404-f005:**
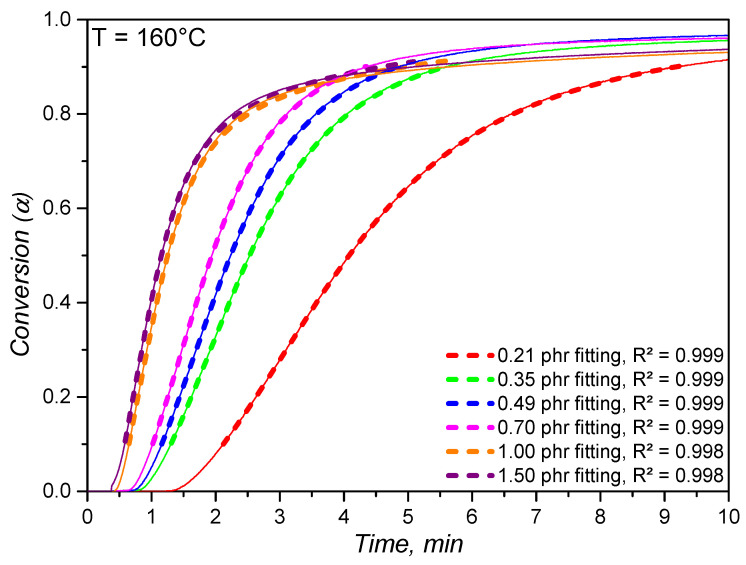
Comparison between the experimental and the calculated conversion values for the crosslinking at 160 ${}^{\circ }$C for silicone rubber compounds with dicumylperoxide concentrations. The calculated conversion values are plotted from 0.1 to 0.9 of conversion.

**Table 1 polymers-14-04404-t001:** Curing parameters for silicone rubber compounds concerning minimum torque (ML) and induction or scorch time ($t_{s1}$) for different crosslinking temperatures and varied dicumylperoxide concentrations.

Temperature	140 ${}^{\circ }$C		150 ${}^{\circ }$C		160 ${}^{\circ }$C		170 ${}^{\circ }$C		180 ${}^{\circ }$C	
	ML	$t_{s1}$	ML	$t_{s1}$	ML	$t_{s1}$	ML	$t_{s1}$	ML	$t_{s1}$
	**(dN m)**	**(min)**	**(dN m)**	**(min)**	**(dN m)**	**(min)**	**(dN m)**	**(min)**	**(dN m)**	**(min)**
0 phr	0.58	n/a	0.57	n/a	0.56	n/a	0.54	n/a	0.41	n/a
0.21 phr	0.61	19.98	0.60	6.16	0.58	2.12	0.56	0.96	0.53	0.53
0.35 phr	0.62	11.13	0.62	3.63	0.58	1.29	0.55	0.64	0.55	0.39
0.49 phr	0.60	9.40	0.59	3.02	0.57	1.16	0.55	0.58	0.54	0.36
0.70 phr	0.61	8.06	0.60	2.63	0.58	1.01	0.56	0.53	0.54	0.35
1.00 phr	0.61	3.76	0.58	1.31	0.56	0.63	0.55	0.57	0.56	0.28
1.50 phr	0.58	2.62	0.56	1.05	0.53	0.57	0.57	0.37	0.53	0.28

**Table 2 polymers-14-04404-t002:** Curing parameters for silicone rubber compounds concerning maximum torque (MH) and optimum cure time ($t_{90}$) for different crosslinking temperatures and varied dicumylperoxide concentrations.

Temperature	140 ${}^{\circ }$C		150 ${}^{\circ }$C		160 ${}^{\circ }$C		170 ${}^{\circ }$C		180 ${}^{\circ }$C	
	MH	$t_{90}$	MH	$t_{90}$	MH	$t_{90}$	MH	$t_{90}$	MH	$t_{90}$
	**(dN m)**	**(min)**	**(dN m)**	**(min)**	**(dN m)**	**(min)**	**(dN m)**	**(min)**	**(dN m)**	**(min)**
0 phr	0.72	n/a	0.77	n/a	0.91	n/a	1.07	n/a	1.28	n/a
0.21 phr	7.76	45.45	9.89	20.92	10.48	8.91	11.25	4.54	11.54	2.92
0.35 phr	8.72	31.15	10.26	12.94	10.86	5.43	11.57	2.99	11.76	1.64
0.49 phr	8.93	27.07	9.97	10.18	10.87	4.72	11.57	2.44	11.64	1.25
0.70 phr	9.4	24.37	9.95	8.93	10.98	4.21	11.57	2.13	11.82	1.19
1.00 phr	11.40	16.74	11.92	8.75	12.39	4.5	12.56	2.93	12.63	1.22
1.50 phr	11.05	12.58	11.55	6.97	11.78	4.14	12.07	4.53	12.06	1.04

**Table 3 polymers-14-04404-t003:** Kinetic parameters determined after fitting of the experimental conversion rate $\frac{d\alpha }{dt}$ to the Kamal model (Equation (Equation 10)) for silicone rubber compounds crosslinked at 160 ${}^{\circ }$C with different dicumylperoxide concentrations.

	$A_{1}$	$A_{2}$	$E_{\alpha ,1}$	$E_{\alpha ,2}$	*m*	*n*	$k_{1}$	$k_{2}$
Sample	(min${}^{-1}$)	(min${}^{-1}$)	(kJ mol${}^{-1}$)	(kJ mol${}^{-1}$)	−	−	(min${}^{-1}$)	(min${}^{-1}$)
0.21 phr	$1.1\times 10^{15}$	$6.4\times 10^{15}$	131.35	131.35	1.167	1.617	0.1591	0.9261
0.25 phr	$6.4\times 10^{14}$	$3.8\times 10^{15}$	127.47	127.47	1.298	1.621	0.2720	1.6150
0.49 phr	$4.4\times 10^{14}$	$2.4\times 10^{15}$	125.75	125.75	1.210	1.552	0.3015	1.6444
0.70 phr	$3.4\times 10^{14}$	$1.9\times 10^{15}$	123.92	123.92	1.350	1.667	0.3872	2.1638
1.00 phr	$3.2\times 10^{10}$	$4.2\times 10^{11}$	87.67	87.60	2.048	2.760	0.8715	11.438
1.50 phr	$1.0\times 10^{10}$	$1.1\times 10^{11}$	83.61	83.61	1.820	2.610	0.8269	9.0960

## Data Availability

All data presented in this publication are only available upon request to the corresponding author, assuming a formal approval of the involved companies and co-authors.

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
