# Peer review of "Peroxide-Based Crosslinking of Solid Silicone Rubber, Part I: Insights into the Influence of Dicumylperoxide Concentration on the Curing Kinetics and Thermodynamics Determined by a Rheological Approach"

_polymers, 2022, doi:10.3390/polym14204404_

Round 1

Reviewer 1 Report

The manuscript presents ax extended study on the cross-linking process of solid silicone rubber based on the concentration of dicumylperoxide. The process is investigated both by collecting experimental data through a rubber process analyser to study the cross-linking kinetics and by applying a theoretical model to describe the process variations using kinetic and thermodynamic explanations. By doing this, they demonstrate that curing parameters (as optimal curing time, scorch time and torques) are affected by changes in the dicumylperoxide concentrations and curing temperature. Besides, the detected curing restrictions observed at high DCP concentrations were correlated to the reaction thermodynamics and to the proton abstraction mechanism explained in the third section of the results. 

Overall, the manuscript is well written and clearly discussed in all sections. The introduction provides a well structured background. The materials and methods section is substantial as it provides all the experimental details and the theoretical equations that are used to discuss the results section. The results are presented in a logical way and drive the reader to the understanding of the curing process in the tested experimental conditions. For this reason I do think that the paper should be considered for publication in Polymers. Moreover, the manuscript deserves only minor revisions before publication. 

In the introduction some details about the current methods used for curing rubbers could be added, as only the necessity to cure silicone rubber is reported without any reference to the current curing methods. Some references should be added in this part of the introduction.

In section 2.2 the authors decided to cure the PDMS/DCP compounds for 1h. Why they decided to use this curing time? Connected to this question, I was wondering if the curing curves reported in Figure 1 could be cut at 40 minutes to facilitate the reader and make the differences between curing parameters more evident from the figure. I did appreciate Figure 2 and 3 as they give a clear picture of the curing parameters, but figure 1 could probably be improved, as all the differences are in the first 20 minutes. 

Author Response

Please see the attachment. Comments are discussed under the section "Reviewer #1, round 1"

Reviewer 2 Report

The manuscript under reweaving is devoted to predicting the optimal curing time at various temperatures of polydimethylsiloxanes (PDMS) by peroxide reagent – dicumylperoxide (DCP). Authors combined two main approaches: the isoconversional principle and the autocatalytic model, that allowed them to suggest the kinetic equation for the rate of conversion at two different temperature profiles using rheological characteristics: minimum and maximum torques. For choosing the equation parameters, authors have considered the chemical aspects of DCP decomposition consisting in formation of the cumyloxy and the methyl radicals, responsible for silicone crosslinking and estimated the activation energies of these processes. Based on NMR data about absence in PDMS vinyl groups, authors devoted the main attention to the proton abstraction from methyl groups. This simplified the equation and allowed constructing the dependence of optimum cure time as a function of the DCP concentration and the temperature.

It seems to me that all parts of the manuscript: physical-chemical, chemical, rheological and applied are written very qualified. Only one question remained non-understandable, namely the kinetics of PDMS curing in injection molding process. How to control curing process during filling the mold?   

Author Response

Please see the attachment. Comments are discussed under the section "Reviewer #2, round 1"

Reviewer 3 Report

This manuscript presents results from a study of curing of silicone rubber with a peroxide. Experimental data are collected from and RPA and a kinetic model is fitted to data using non-linear regression techniques in order to estimate model parameters. Although the approach is not new or novel, the manuscript deserves publication with some minor revisions as it presents data on a new silicone rubber and it is very well written. Please address the following comments in the revision.

1.      It is stated that the rate of cure is plotted against temperature in Figure 3b. However, Figure 3b shows the rate of cure as function of conversion. Please revise this statement or adjust Figure 3b.

2.      RPA data are fitted to equation (10). However, both the dependent variable (rate of cure/ conversion rate) and the independent one (conversion) are calculated from the same confounding variable (torque). It would have been more appropriate to separate dependent (conversion) and independent (time) variables in equation (10), integrate and then do the fitting. The authors need to comment on this and on the validity of the model parameter estimates.

3.      Data in Figure 4b and Table 3 show a reduction of activation energy with conversion. True activation energies do not depend on conversion. This indicates that other phenomena are significant at higher peroxide concentrations. The authors comment in this reduction of activation energy on page 12 but need to expand the discussion to address heat/mass transfer limitations.

4.      In lines 389-392, the authors discuss a hypothesis. This is not a hypothesis as it is well known that the crosslinking efficiency of dicumyl peroxide is reduced at higher concentrations. The authors need to include some references to that effect (e.g. JOURNAL OF POLYMER SCIENCE VOL. L, PAGES 287-298 (1961))

5.      Figure 5 does not add any value to the manuscript since the model parameters were estimated by fitting the model to the same data!

6.      Appendix A1: equations A1 can be removed since they are well known to anyone working in the field. Also, Figure A1 can be eliminated since the same information is presented in Table A1.

Author Response

Please see the attachment. Comments are discussed under the section "Reviewer #3, round 1"
